# The MOVING GROUND Project: A Nature-Positive Case Study

**Nicholas Anastasopoulos** [1,*]**, Penelope Iliaskou** [2] **and Mariela Nestora** [3]

1 School of Architecture, National Technical University of Athens, 15780 Zografou, Greece
2 Duncan Dance Research Center, 16232 Vyronas, Greece
3 ArtEZ Institute of the Arts, Onderlangs 9, 6812 Arnhem, The Netherlands
* Correspondence: nanastasopoulos@arch.ntua.gr; Tel.: +30210-6918735

**Abstract:** This paper is a report on the year-long MOVING GROUND project (MG), initiated by the Isadora and Raymond Duncan Dance Research Center (DDRC). The Duncan Dance Research Center sets out to address climate change issues interweaving the social, physical, and artistic spheres by introducing the concept of a garden both literally and metaphorically to inspire the artistic community and shift mindsets of the local community. By a gradual transformation of its grounds, infrastructure, and social fabric, the long-term goal of the DDRC is to function as a tangible model that can be experienced and replicated as a whole or in parts in the city or elsewhere. The paper discusses the goals, methodologies and strategies introduced during the project aiming towards the regenerative transformation of the institution that drew inspiration from permaculture principles, nature-based solutions and a net-positive design perspective. The paper also discusses the novel experimentation of applying permaculture principles to artistic creation and practices. The paper concludes with a reflection of the outcomes and an assessment of the goals it set out to achieve.

**Keywords:** net-positive design; nature-positive development; regenerative design; degrowth; nature-based solutions

## 1. Background

The Isadora and Raymond Duncan Dance Research Center (DDRC) is a historic place that belongs to the municipality of Byron and is part of the greater metropolitan conglomerate of Athens, Greece. It was founded in 1903 by Isadora Duncan and her brother Raymond Duncan. The classical Greek civilization had a profound influence on the Duncans. Isadora Duncan is widely recognized as the founder of modern dance due to her then radical departure from the conventional ballet tradition [1]. The site where the DDRC is located was chosen for its then commanding view of the Acropolis and the Saronic gulf. The building's austere characteristics, which Raymond adopted, were said to have drawn inspiration from Agamemnon's Palace in Mycenae. The area surrounding the DDRC belongs to three municipalities (Byron, Hymettus, and Athens), all of which followed the typical Athenian development model that roughly took place between 1960 and the late 90s, characterized by incremental, privately funded residential construction with the absence of urban planning. Public infrastructure customarily arrived too late, only to provide basic services to new construction in neighborhoods by providing access to water, electricity grid and sewage networks. During the past 120 years, due to urban growth around the center of the city, the 16,000 m² of Duncan grounds were reduced to 1500 m², and the multi-story apartment buildings, which continued to replace one- or two-story houses until 2010, obstructed the direct view of the Acropolis. In 1984, the building structure was recognized as a cultural heritage monument of architectural and historic significance. The original edifice where the DDRC is housed today largely retains its original characteristics. After extensive renovations, which occurred in 1986 under the auspices of the Byron municipality, the DDRC became established as a contemporary dance

center, offering services to both the professional dance community as well as the local community. The DDRC forms part of the European Dancehouse Network (EDN).

MOVING GROUND (MG) was an experimental interdisciplinary project initiated by the DDRC and funded by the Greek Ministry of Culture. The overarching goal of MOVING GROUND was to foster ecological awareness, cultivate active citizenship, and address the microclimate of the area around the DDRC through a series of short-, medium-, and long-term interventions. In this framework, MG was designed as a research hypothesis that would examine the changes reflected at the DDRC facilities, its grounds, and the communities using it, as a result of its methodology which is discussed here. It would address the dance community, as well as the local community, through a series of activities hosted at the Duncan Dance Center and at adjacent locations testing the potential impact of dance and garden care on the quality of life.

The overall concept and design of the MG project was coordinated by the artistic director of the Duncan Center and was undertaken by an interdisciplinary team consisting of two choreographers, one visual artist, two architects, and one film director who was also a member of the local community. The curatorial team adopted a direction that saw the human factor as the main driving force towards any change and adaptation strategies. Throughout its duration, MG involved local people, dancers, choreographers, students, architects, researchers, artists, botanists, permaculture educators, scientists, teachers, as well as citizens of all ages from the community of the Municipality of Byron and beyond. Besides choreography projects, it also included educational workshops and a series of events of ecological nature and content, all of which were introduced and developed in dialog with the design and implementation of a local community garden. During the first stage of the project, it was sought to identify and document the typology of its grounds and the landscape by exploring nature's cycles, and by highlighting nature's presence in an urban setting. To this end, a series of actions was initiated that aimed to regenerate the depleted ecosystem around the center by focusing on the soil, flora, the characteristics of the landscape, and the overall environment. At the same time, the residents' involvement and active participation was encouraged through a series of collective actions that addressed the quality of life of the local community and fostered active relationships of cooperation.

Raymond Duncan is seen today as a visionary who questioned the impacts of industrial civilization. Trying to put his ideas into practice, he adopted a lifestyle of frugality, probably influenced by similar movements occurring elsewhere in the US and Europe. He was an advocate of self-limits and human labor versus mass production, principles he abided by throughout his life [2]. These principles resonate today as an inspiration for rethinking the built environment, the orientation of the contemporary practices of artistic production, and Western civilization at large. Paying tribute to its founder through the MG project, the DDRC addresses, once again, community development, active participation, green transition, and frugal living.

Athens, which is the setting of the DDRC, is a city that is especially vulnerable to the effects of the climate crisis due to its location and the characteristics of the way that it has developed, experiencing severe heat island effects, prolonged heat waves during the summer, torrential rains and flooding with increasing frequency, and extreme weather phenomena. An overarching goal of the MG project in all its phases has been to address this urgent topic through a holistic approach in the series of interventions adopted [3].

## 2. General Principles

As its main goal, this project set out to explore the main concepts and principles of Permaculture and apply them to different layers that included the social fabric of the community and the physical grounds of the DDRC (Table 1). The three fundamental Permaculture ethical principles of "*Earth care, People care, and Share the Surplus*", together with the twelve principles of practice [4,5], functioned as underlying presets of the MOVING GROUND project supporting regenerative aspects of design, and emerging behavioral patterns of positive impact. In addition, the project tackled sustainable development goals

11/12/13 and 15, specifically, sustainable cities (11); sustainable consumption and production patterns (12); action to combat climate change and its impacts (13); and the protection, restoration, and promotion of the sustainable use of terrestrial ecosystems, combat decertification, reversing land degradation, and halting biodiversity loss (15). However, in line with positive development theory, MG viewed the SDG targets and indicators as being often mild, ineffective, and/or contradictory, as they are based on 20th century contested economic indicators like GDP. Positive Development theory is grounded in the conviction that human constructs, especially through physical and institutional design, can do far more than reduce harm caused by urban development or achieve zero energy: they can and must reduce planetary overshoot by increasing nature, ecosystems and biodiversity, by avoiding pollution, reversing climate change, building community, and by increasing social justice [6,7]. The MG project strived to highlight these shortcomings by adopting a net-positive development approach of giving back to nature and the community more than it takes, in both whole system (space) and life cycle (time) terms.

**Table 1.** Permaculture principles applied in various events and observed results.

| Event | Applied Principle | Results |
|---|---|---|
| | Use and value diversity | |
| See Section 5 Milestone actions and events | The entire MG program was conceived through a transdisciplinary program and events which included multiple audiences and components. | Artists and residents of all ages developed strong community bonds, reflected in their desire to continue past the end of MG |
| | Observe and interact | |
| Flora identification and arboretum creation (Manos Bazanis, Nikos Valcanos and Vangelis Skoufakis) | Actions that identified and documented the typology of the ground and the landscape | Introduced the ecosystem of the surrounding site and the flora it hosts. Made visible to participants nature's practices in an urban setting |
| | Design from pattern to details | |
| Plant nursery workshop,Nature-based solutions lecture,community composting unit | A pattern of weekly meetings of the curatorial team and monthly meetings with participating artists was maintained. Consistent presence through the steady rhythm of organizing monthly public. | The year-long MG project's activities were tuned with the four seasons, organizing accordingly works at the garden. |
| | Accept feedback and creatively respond to change | |
| Regular meetings between the curatorial team members, and consultation meetings with the artists | The participant artists creatively responded to change, often through using the feedback methods proposed by the curatorial team. | Adjusting and becoming more resilient to change |
| | Produce no waste | |
| Compost-making and compost bin construction | All public events were implemented with a no waste policy. Residents have become familiar with the composting process and regularly use it. | Compost is being used in the garden |

Through a series of interventions occurring in whole system and life cycle terms, MOVING GROUND attempts to address the physical grounds at various stages (short, mid, and long term), recognizing the time, money, labor, and other resources required, but, also, because these interventions are expected to organically emerge from a process of gradual transformation of the mindset of participants. Nature-based Solutions (NbS) which are defined as: "Actions to protect, sustainably manage and restore natural or modified ecosystems that address societal challenges effectively and adaptively, simultaneously providing human well-being and biodiversity benefits" were also considered as strategies

specifically helpful when applied on urban centers that could contribute to adapting and mitigating the effects of climate change [8,9]. The NbS toolkit was taken into consideration for midterm and long-term strategies to address water scarcity and security, disaster risk, and desertification.

## 3. Methodology

MOVING GROUND was framed by four interconnected pillars: dance, space, community, education. These four pillars were addressed and developed equally and in parallel. Their combination was meant to encourage rhizomatic growth, and to embed sustainability in practice.

As the disastrous effects of climate change increase in frequency and intensity, and the need for relevant action becomes more urgent, in line with many institutions around the world, the DDRC chose to challenge artists to come up with ideas, projects, and activities that raise ecological awareness, thus, tuning the art scene to the urgent issues of the present. Artists of a wide age, experience, and artistic practice range were invited to think through their needs, interests, and energy consumption patterns in relation to the four pillars of MG in order to generate proposals for artistic and educational programs. A gradual regeneration process of the derelict grounds was initiated with the clearing of garbage and debris which were used for the creation of a mandala, followed by the initial steps of creating a community garden. The residents were invited and were encouraged to participate in several workshops and events. Finally, experts were invited to offer workshops and talks in order to engage the public, and courses were designed to address several ages and needs.

Various activities have already taken place, primarily those being relatively easy and requiring little or no funding, addressing the importance of diversity, cooperation, interdependence, observation, and care. They formed part of the short-term strategy which coincides with the duration of one year of the MOVING GROUND project. The year-long MG's goals addressed engaging with the communities and the initial steps of regenerating the immediate surroundings of the DDRC, and the MG strategy will be succeeded by a three-year mid-term plan, and a six to ten years long-term plan. The long-term goal is the gradual shift of the DDRC's operations towards artistic practices and infrastructural interventions that are regenerative, transformative, and in tune with both the urgent environmental, social, and political issues of our times and the specificities of the Athenian urban condition.

Here, the four MOVING GROUND pillars are explained in detail:

a.   *Designing a Garden*

The 18th century engravings of the Acropolis and the classical monuments of Athens depict an arid and bare landscape, which extended across the horizon for as far as the eye could see. Kopanas hill was chosen by Raymond Duncan for its advantage of a direct visual relationship to the Acropolis; nevertheless, this advantage was overshadowed by the discovery that came soon after. The site had no access to water. Aridness was eventually a determining factor that made the self-sustainable community that Raymond envisioned unsustainable. Later on, after the place was deserted and as the area was rapidly being populated, in 1935, the public water company of Greece created a water reservoir on the piece of land adjacent to the Duncan Center to respond to the rising water supply needs. To this day, some of the elements which characterize Kopanas hill are aridness, erosion, and depleted topsoil of what remains of the slope, which, due to rapid urbanization, has been fragmented into pieces of land belonging to different entities and cut by roads. These are elements that often form the adverse circumstances that characterize the Mediterranean climate. As a result, the Mediterranean landscape has adapted to a distinct ecosystem that is composed of flora that is capable of surviving in poor soil and has adapted to the dry months of the summer (Figure 1a,b). The physical challenge has been to transform the desolate and neglected grounds around the Duncan Center, which was caught in a downward spiral of neglect that encouraged vandalizing acts by the same people who

used the public space's surroundings. Among other things, the degraded surroundings suffer from fragmentation because of property division and thoughtlessly giving priority to cars over pedestrian access, resulting in compromising the integrity of the hill. The first step taken was succeeding in getting permission to use the adjacent enclosed area of the disused water reservoir, which allowed its transformation into a local communal garden (Figure 2a,b). The design, cultivation, and maintenance of the community garden is understood as a communal activity embodied through care, the adoption of a habit, and collaboration. Thus, the ecosystem was used as a metaphor for the flourishing and thriving of artistic practices as well. In the light of the need to become more conscious of interconnectedness, evolution, and ecology, MG assembled relationships between artistic practices and the concept of a garden, which were articulated in various projects. Through a series of actions, a culture of sharing and commoning has been introduced.

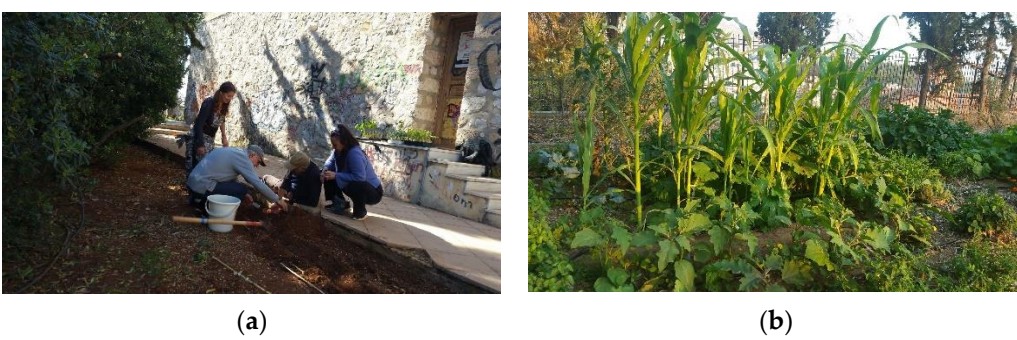

(**a**)             (**b**)

**Figure 1.** Preparing the garden beds and yielding the first crop. (Photos credits: Nicholas Anastasopoulos (**a**) and Monica Vaxevani (**b**)).

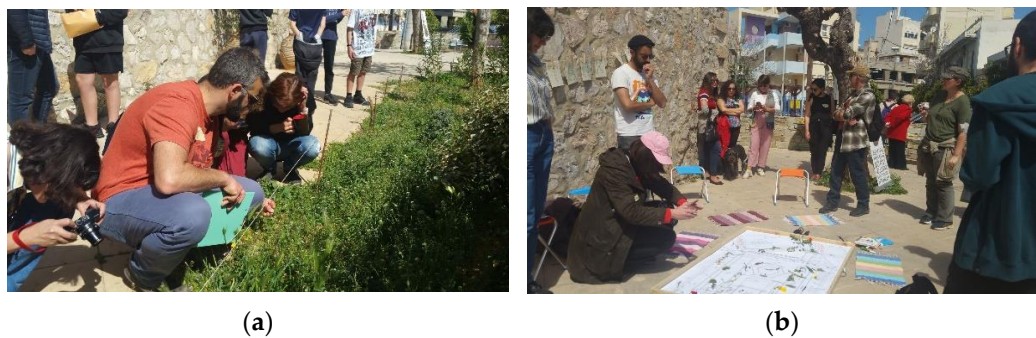

(**a**)             (**b**)

**Figure 2.** Identifying plants (**a**) and making an arboretum (**b**) (Photo credits: Monica Vaxevani).

The garden was primarily based on two principles and two different schools of thought. The first school of thought recognizes the harshness of the Mediterranean climate and the local microclimate, which has resulted in particular families of indigenous plants adapting to approximately six months of rain and six months of dryness, during which many plants go into summer hibernation [10]. The second school of thought involves the introduction of irrigation for the making and maintenance of an aesthetic and, at the same time productive garden by incorporating small scale food production, experiential involvement, and training. For this purpose, rare heirloom varieties have been introduced from seed banks, which are generally known for their adaptiveness and resilience to pests [11]. For many people, a dry garden is not a particularly pleasant sight to see, and it takes a change of mindset and educating public and landscape design professionals regarding biodiversity and water scarcity so as to incorporate and celebrate this condition.

*b.*     *Developing Sustainable Artistic and Dance Practices*

MG invited projects that would be intertwined with the design, making, care, and maintenance of a communal garden, and biodiversity was treated metaphorically as a char-

acteristic of artistic creation. The artist was addressed as an agent who is just as involved in building knowledge and adaptiveness, as the environment or the plants themselves. The challenge posed to participating artists was to rethink dance as a form of artistic creation capable of inspiring change in social and ecological attitudes, and to examine the type of knowledge that might be generated through such cross-pollination guided not solely by rationality, but also by speculation and radical imagination. (Figure 3a,b). As a result of this approach, a highly diverse interdisciplinary series of works emerged and some of the questions informing the project through dance and choreography were the following:

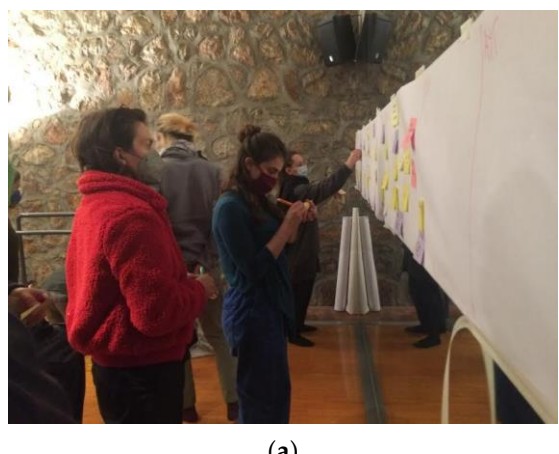

(**a**)

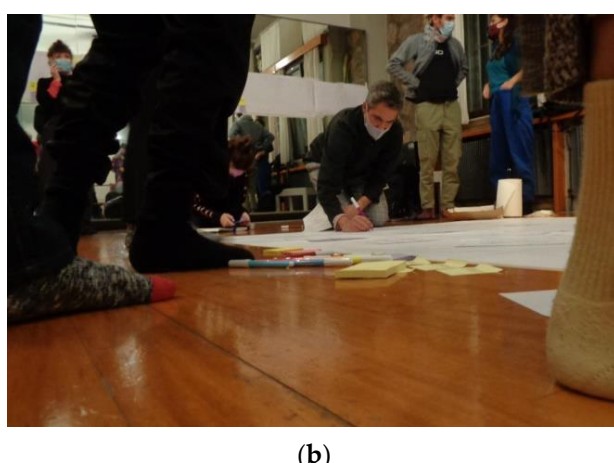

(**b**)

**Figure 3.** Dancers and choreographers that responded to the open call co-produce the calendar of events (**a**) working together in search of complementarities between their proposals (**b**). Event organized by dancer and choreographer Vitoria Kotsialou and Mariela Nestora (Photo credit: Monica Vaxevani).

How can we permeate dance with permaculture theory and practice, and, conversely, how can we inform permaculture practice through dance? How can dance involve the community, connect with ecological concerns, send powerful messages, and bridge the distance between art and life? Which dimension of a garden would be most meaningful to include in the ways that we organize group or office work, dance communities, research, choreographic projects, communities, or educational projects?

The main guidelines specifically proposed to participating artists were:

1. To design proposals conceptualized through the filters of Permaculture principles. Some included, for example, replacing the notion of growth and progress with the word "yield", efficiency with "resilience", or programming with "pattern-setting"; Thus, the individual projects were reformulated and re-articulated as patterns;
2. To interconnect their respective projects through weekly meetings at the DDCR, to co-work in parallel, to collaborate, and to participate in monthly collective feedback meetings between the curatorial team and participating groups;
3. To engage in a culture of support of each other's projects by hands-on collaboration, feedback, and observation;
4. To lead or co-lead one of the six "Permaculture and dance workshops" addressed to professional dancers;
5. To join in co-organizing and contributing to a monthly event series titled "Gathering in the garden", in line with the season and the garden needs;
6. To imagine ways in which their projects can involve the local community, and different age groups, in line with the other educational projects. As quality of life in a community is often dictated by the degree of engagement and the joy that individuals can draw from communal interactions. MG aimed to inspire group connections by building relationships between community members through activities and hands-on activities involved in conceiving and making a garden together;

7. To embrace the notion of audience engagement by designing artistic projects in which audience members were invited to participate, rather than being mere observers or consumers of experiences and knowledge; to contribute to short-, mid-, and long-term perspectives and curatorial tenets;

8. To participate in creating an open MG library where articles, books, and references were shared amongst participants.

*c.* *Expanding Forms of Education, Artistic Practices, and Communication*

The series of educational programs that was established through MG catered to different age groups. Workshops on dance, creative movement, and visual arts addressed kids (preschool and primary school ages 4–7). A series of environmental and educational actions addressed and actively involved high school adolescents (ages 12–17) from the public schools of the municipality of Byron and were based on the fundamental principles of experiential methods and active participation in the learning process. In addition, a gentle dance practice enhancing ease and creativity addressed retired people and a third age group (ages 65+). Attempting to enrich the feedback process and communication patterns, a visual artist was invited to make "response art" during the artistic and community project activities. Coming from the art therapy context, Response art enriched the feedback process, communication patterns, and provided an aesthetic awareness in the act of witnessing and responding to the projects. The kick-off event, a mandala produced by the waste collected from the plot, was transformed into a visual response and aesthetic experience (Figure 4a,b). Response art was consequently exhibited in the space providing an aesthetic experience and appreciation of the activities. [12].

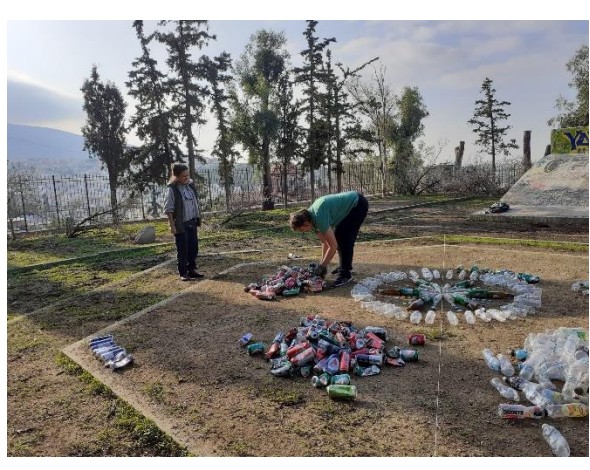
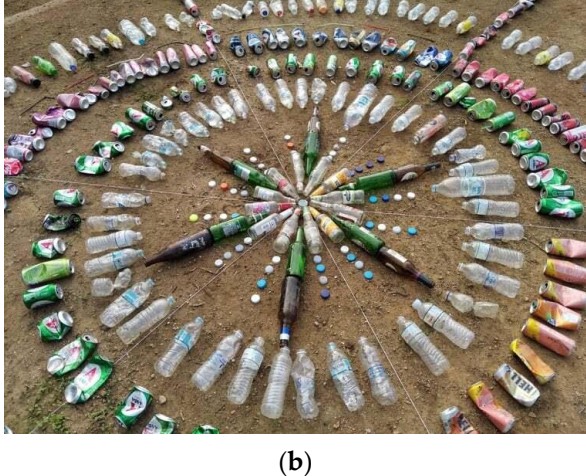

(**a**)　　　　　　　　　　　　　　　　　(**b**)

**Figure 4.** Communication artist Christine Katsari creating a mandala with trash collected on the site (**a**) and the mandala completed (**b**) (Photo credit: Lena Kita).

*d.* *Building Community*

One of the challenges for the MOVING GROUND project has been communicating its vision to the broader community, attracting more people, and forging trust and a collective imaginary in creating a sustainable environment together. The power of community involvement is well recognized by scholars, as it has been shown to help combat the sense of isolation that may be felt in urban settings, especially among social minorities. Literature shows that committing to collective, meaningful causes, which involve social interactions such as working together or volunteering, strengthens social bonds, enhances a sense of purpose, and has the potential to take roots and to grow [13]. The relationships between people and spaces have been called *relationscapes,* alluding to the power of relationships which populate a locality and make it sustainable and resilient [14,15]. Community empowerment may impact an urban environment in multiple ways by helping people through difficult times, when in need of help and encouragement, producing bonds and resilience,

resulting in active participation (Figure 5a,b). Greece was hit by various crises in recent years. In 2009, a social and political crisis erupted which was succeeded by the economic crisis of 2011, followed by the refugee crisis of 2015, and the coronavirus pandemic of 2020. Since 2009, several collectivities and initiatives were born addressing all aspects of life. Nevertheless, a deep-rooted mistrust towards common causes and institutions, largely endemic to Greek culture, constitutes a challenge in sustaining community involvement. People are more likely to take care of their own backyard or balcony—even the pavement in front of their door—but will not easily cross the threshold from the private to the public domains, even more so the commons. Positive community experiences allow individuals to have a sense of belonging and better connection to their environment and other people.

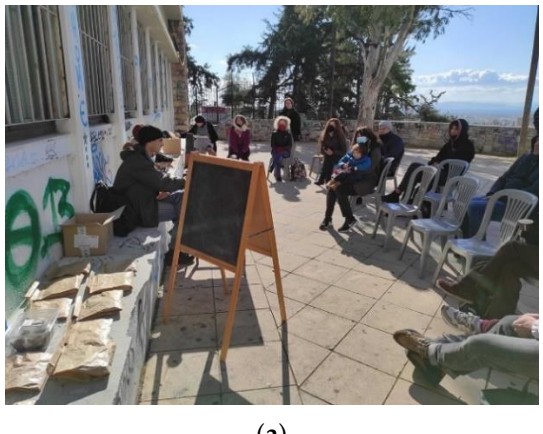
(**a**)

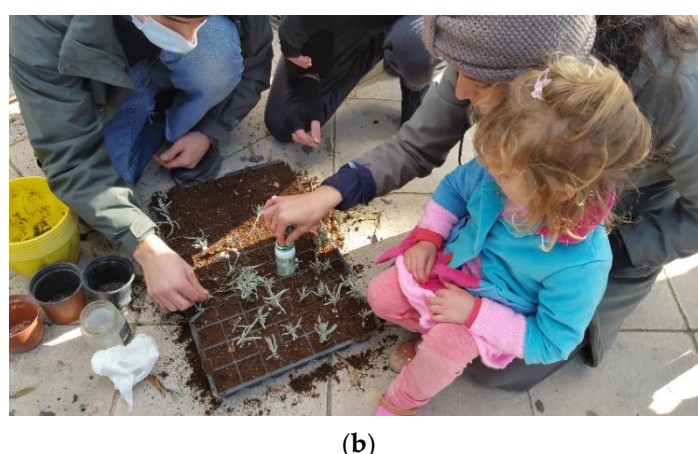
(**b**)

**Figure 5.** A seed exchange and germinating techniques seminar at the terrace (**a**) and planting seedlings by the workshop participants (**b**), 23 January 2021. (Photo credit: Monica Vaxevani).

## 4. Milestone Actions and Events

A multitude of projects and events unfolded during the twelve-month period of the MG project. Some of the most relevant events which outlined the methodology implemented, were the following:

A Permaculture workshop (theory and practice taught by Tina Lymperis). The workshop disseminated permaculture principles and involved participants in planting seasonally appropriate nasturtiums (Tropaeolum majus) around the Duncan center as a first gesture of common care and collective thinking.

A Permaculture talk (taught by Bram van Oberbeeke). The talk offered ways of thinking by applying permaculture principles and the permaculture approach into groups or collectives around themes of working together and effective organizing.

A lecture on Nature-based solutions by Tannya Pico Parra, an Ecuadorian architect, and researcher, PhD candidate on Urban Development and Governance at the Institute for Housing and Urban Development Studies of Erasmus University Rotterdam offered insights into relatively simple approaches which involve natural processes towards making more resilient and user-friendly urban settings.

A Flora identification and arboretum project by Manos Bazanis, Nikos Valcanos, and Vangelis Skoufakis addressed the characteristics of urban plants which are normally invisible and ignored as weeds, and nature's adaptive processes were revealed.

The workshop of building a communal composting unit by the Earth Organization NGO introduced ways of communal making of compost which can be used locally, and can be fed and maintained by the local inhabitants. The compost produced is being used already in the garden, and can benefit community members, and the municipality.

A plant nursery workshop (organized by Vasilis Ntouros and Dora Zoumba) workshop introduced seeds and tehchniques of growing plants from seeds to inhabitants. Workshop participants were encouraged to take this knowledge of plant growing from seeds home, and to later return some of these home-grown plants to the newly established community

garden. Most of the activities involved the exterior space of the DDRC, in order to establish it as public space and as a common good inviting encounters, reflections, expressions, and interactions.

## 5. Results and Further Reflections

The MOVING GROUND project closed its first-year cycle in September 2022, and in November of the same year a celebration was held which offered the opportunity of assessing the outcomes.

The learning outcomes and experiences were many and diverse. Just as diverse were the artists, participants, and audiences involved. During the twelve-month period of MG, the project adopted a program of actions that observed the cycle of the seasons. The first steps were an exercise in reaching out, cooperating with each other, overcoming conflict and managing crises, becoming resourceful, and negotiating access to resources (land, water, and materials) with various stakeholders that included the municipality and public water company, among others. The dance community's horizons of artistic creation were expanded with new forms of creation, resulting in hybrid projects that included dance workshops, gatherings, public talks, promenades, hands-on craftsmanship, making, and gardening. These experiments constitute a novel contribution to the ongoing debate about how artistic practices may better respond to the environmental challenges of biodiversity loss and the climate crisis.

Numerous educational programs catering to different age groups were implemented with schools of neighboring municipalities and the local community, which raised awareness regarding responsible practices about limited resources, and water management. Through educational events imparting theoretical knowledge, as well as hands-on activities, the center disseminated the principles of permaculture, sustainability, net-positive ethics, and nature-based solutions to diverse audiences. The physical grounds of the DDRC experienced a gradual, and palpable transformation around the creation, use, and maintenance of a communal garden. The unique spirit of the place (genius loci) was strengthened and long-standing behavioral patterns and habits were altered. Furthermore, in the following months after the herbal and vegetable gardens were planted, a rapid increase in biodiversity was observed, as the space has been attracting a wealth of insects and birdlife, now offering visual and olfactory pleasures. Additionally, physical changes which begin reversing the climate of a desolate urban void into a much-needed, welcoming open space that offers a respite from the often-harsh climate of Athenian urban life are noticeable.

People of all ages and walks of life became motivated to participate in the maintenance and care of the newly established communal garden. This involvement may have encouraged them to think about practicing similar methods in their immediate surroundings. Intricate relationships between the personal, the social, and the environmental spheres were forged through the experience of simultaneously observing and choreographing actions addressing the four pillars of the multidisciplinary MG project. The local community became aware of sustainable daily habits and was encouraged towards adopting a mindset of belonging, active involvement, and conscious consumption, introducing concepts of degrowth in line with Raymond Duncan's ideals. The connections established between the physical and corporeal aspects of dance practices through the introduction of permaculture opened new territories of knowledge and allowed a different set of ethics to develop.

During the period that the project took place, the DDRC was host of the EDN Atelier (Figure 6), an event welcoming the EDN European Network (1 and 2 June 2022). The event was designed with the aim to inspire participants to rethink their own artistic and daily practices in dance centers and communities throughout Europe, through an extended process of unconventional sharing and learning that disseminated the MG learning outcomes [16]. During the time that the project was active, the Duncan Center was used as a case study in two courses at the School of Architecture at the National Technical University of Athens, resulting in several student projects. These academic activities allowed several debates around climate change, community participation, permaculture, nature-based solu-

tions, and net-positive design to occur through fieldwork and bibliographical research [17].

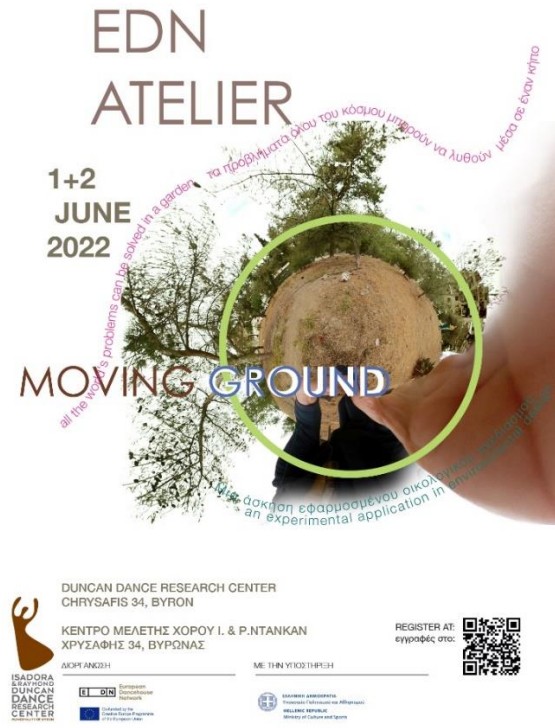

**Figure 6.** Poster for the EDN Atelier two-day event, 1 and 2 June 2022. Design: Dimitris Theodoropoulos.

Mid-term plans involve addressing the management of the public land outside of the confines of the DDRC through the conceptualization and reconfiguration of the surrounding areas into an integrated landscape design and of the water reservoir. These strategic moves will require cooperation with other public and private entities such as landowners and public authorities and stakeholders. More specifically, mid-term plans include further enhancing the space of the abandoned reservoir which has become the communal garden through the introduction of new zones which include the creation of a shade-offering pavilion for social gathering and performances, and the configuration and design of an observation point for urban gazing. Addressing the current boundaries that compromise the integrity of the hill is also a challenge and major priority. Long-term plans include rainwater harvesting for the needs of the garden through the reuse of the empty reservoir as storage, by activating and redesigning the roof and exterior paved surfaces of the DDRC. The introduction of a grey water reuse system from the daily use of the facilities of the center is also foreseen. Within a mid-term and a long-term range, issues of energy transition and the current model of development will also be addressed [18,19]. Taking this into account, the curatorial team has been acting toward securing funds for another year of activities, and has applied for European funding through a collaboration with projects facing similar challenges throughout the South of Europe and the Mediterranean.

The Duncan Center's capacity to provide conditions that inspire trust and participation of the local community and to support innovative creative work was reinstated. The sustainability of what has been achieved will be judged by the consistency and how the practices established will be observed and further developed in the future.

**Author Contributions:** Conceptualization, N.A. and P.I.; Validation, M.N. All authors have read and agreed to the published version of the manuscript.

**Funding:** This research received no external funding. The MG project was funded by a seed fund of the Greek Ministry of Culture.

**Acknowledgments:** The MOVING GROUND curatorial team is composed of: Penelope Iliaskou., DDRC Director. Mariela Nestora, choreographer. Vitoria, Kotsalou, dancer/choreographer. Dimitris Theodoropoulos, architect, Christina Katsari, community artist, Monika Vaxevani, Film director and community animator, Nicholas Anastasopoulos, architect/professor at NTUA. Fay Zika, Associate Professor of Philosophy and Art Theory at the Department of Art Theory and History of Art at Athens School of Fine Art, Gigi Argyropoulou arts researcher, curator and Elena Gogou, permaculturist, for the inspiring keynote speeches at the EDN Atelier event. To Betina Panagiotara and Anastasio Koukoutas for the insightful report. To MG artists Dora Zoumba, Vasilis Ntouros, Iris Nikolaou, Vasiliki Tsangari, Katerina Delakoura, Mariela Nestora, Zoi Dimitriou, Anastasia Polychronidou, Anastasia Barka, Sonia Ntova, Maria Papadopoulou, Vera Karavakou, Despoina Hatzipavlidou, Anthi Mouriadou, Loukiani Papadaki, Elton Petri, Chara Kotsali, Candy Carra, Yannis Tsigris, Mina Ananiadou, Iro Vassalou, Dimitra Mitropoulou, Dimitra Mertzani, Pavlos Simatis, Stella Tripolitaki, Eugenia Demeglio, Thanos Polymeneas, and each and every participant of the EDN Atelier program, 1 and 2 June 2022. Each and every member of the local community who supported the Moving Ground initiative and continues to do so through the caring of the garden and the participation in voluntary acts of support towards the Duncan Center throughout the years. Additionally, to Tannya Pico Parra, Ecuadorian architect and researcher, who offered a lecture on Nature-based solutions. To the administrative and technical support of the DDRC staff members, and to the donations from the Municipality of Byron.

**Conflicts of Interest:** The authors declare no conflict of interest.

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
