# Peer review of "The MOVING GROUND Project: A Nature-Positive Case Study"

_urbansci, doi:10.3390/urbansci7010019_

Round 1
Reviewer 1 Report
The article depends on description of several events around a building that takes place in Athens. The whole manuscript is descriptive rather than being argumentative or outlining how can these important aspects be achieved through buildings form and design in relation to its surrounding. I am not sure how could the larger audience benefit from it.
I find it majorly lacking the author/s voice in building a scientific article and making use of the information put here. I could not find anywhere what the authors wanted to say at the end! what methodology did they follow, or what was even their main research question. Even the very few references referred to here can barley be counted as proper/ rigor scientific references, they are mainly based on handbooks, videos .. etc!!!
Also, I think the author/s submitted an incomplete manuscript, with lots of formatting mistakes!!
Author Response
Response to Reviewer 1 Comments
The article depends on description of several events around a building that takes place in Athens. The whole manuscript is descriptive rather than being argumentative or outlining how can these important aspects be achieved through buildings form and design in relation to its surrounding. I am not sure how could the larger audience benefit from it.
Response 1: Thank you for your valuable comments. It is true that the paper was less argumentative and more descriptive. The text has been drastically revised both in structure and content. I believe that by restructuring the shape and content we address that to a large extent. Nevertheless, the nature of this paper is a case report and, in our view, four months later after the closure of the project it serves as an assessment. This involves the short-term interventions which did not set out to have any impact at the building level, other than some interior adjustments which are not of interest. Regarding your remark about lack of interventions at the infrastructure level, as it is explained, the year-long Moving Ground project is an introductory stage of regenerative strategies which will be followed by the midterm and long-term ones during which building rehabilitation is going to take place. The paper focuses on interventions that occurred between September 2021 and September 2022 which involve mainly the creation of a garden and the first wave of actions to increase biodiversity, and the motivation and involvement of the local community, as well as a dance community. This is due to the limited financial and human resources, but also due to a conscious decision to devote our energies to creating a strong social foundation of support, an effort which did succeed.
Similar case studies do exist around the world. Yet, to our knowledge this is the first case which proposes a novel intersection between dance practices and environmental regeneration. In our opinion a larger audience may benefit from getting acquainted with its outcomes and by learning about the methodology which was used.
I find it majorly lacking the author/s voice in building a scientific article and making use of the information put here. I could not find anywhere what the authors wanted to say at the end! what methodology did they follow, or what was even their main research question.
Response 2: Through the drastic restructuring and rewriting of the text we believe that your valid comments have been addressed. The text now is more argumentative and has a point of view which largely is the one that was formulated by the curatorial team, the author's being members of it. Please see the newly established Methodology section (4) and the Results and further reflections closing section (6).
Even the very few references referred to here can barley be counted as proper/ rigor scientific references, they are mainly based on handbooks, videos .. etc!!!
Response 3: In response to your remark, references and citations have been substantially revised and extended. Please check the references and see if there are a few major other ones that you see missing and would like to suggest.
Also, I think the author/s submitted an incomplete manuscript, with lots of formatting mistakes!!
Response 4: I believe that substantial work has occurred in the formatting of the revised article. I hope you will agree
Reviewer 2 Report
Dear Authors
Thank you for submitting manuscript.
I have some min questions which are problematic.
What is the purpose of writing this?
How this is the novel contribution in science?
What are the hypothesis and objectives of the manuscript?
Authors must consult a vast literature to write this again.
I am confused that, authors have mentioned this is the article.... Where are proper sections which describe the original article?
Please note that, these comments are not the reflection of the content importance but i am concerned about the manuscript in peer review.
Author Response
Response to Reviewer 2 Comments
What is the purpose of writing this?
How this is the novel contribution in science?
Response 1:
The novelty of this project is the connection it establishes between the widely acknowledged significance of permaculture principles with dance practices, and the attempt it makes to apply them holistically in an entity which forms part of the public sector in the city of Athens. This has been better described in part
What are the hypothesis and objectives of the manuscript?
Response 2:
As discussed in the Introduction section (2), Athens is a city at high risk to be affected by climate change with adverse consequences. The hypothesis described is that a positive example of a holistic nature which addresses both the physical, and as well as the social aspects of our case study has better chances to be successful and therefore to establish itself as an example which can be replicated.
Authors must consult a vast literature to write this again.
Response 3: Point well taken. We put great effort in reviewing the existing literature in the areas of permaculture, nature-based Solutions, and participatory processes. Please check the references and see if there are a few major ones that you see missing and would like to suggest.
I am confused that, authors have mentioned this is the article.... Where are proper sections which describe the original article?
Response 4: The article was misrepresented indeed. In our opinion now it is better categorized as a "Case report". Following your remark, this change of category has already been made.
Please note that, these comments are not the reflection of the content importance but i am concerned about the manuscript in peer review.
Response 5: I do hope that the above answers this concern as well.
Reviewer 3 Report
The paper represents unique research due to the particular project that was presented (Isadora and Raymond Duncan Dance Research Center.). However, there are several changes that are required in order for the manuscript to be considered scientific. The research design, hypotheses, and methodology should be clearly stated in the first section of the paper as well.
The abstract could be restructured for a better understanding of the topic. Starting from the context and significance of the research, followed by the aims and methodology of the research, and concluding with the results and possible contribution of the research...Also, the scientific methodology used for the research should be presented in the abstract, as well as in the text.
The connection between the text and the previous and present similar research should be elaborated more. The paper should consider a deeper background theoretical research in the domain of reconstruction or renewal and regeneration of architectural and cultural heritage, i.e. the projects that raise ecological awareness ...
Figure 3 is not readable, due to the small text size.
There are several blank pages in the text (or part of the pages) pages 5 and 12.
The paper lacks a discussion of the results (related to the previous and present similar studies) as well as the overall conclusion of the research.
Author Response
Response to Reviewer 3 Comments
The paper represents unique research due to the particular project that was presented (Isadora and Raymond Duncan Dance Research Center.). However, there are several changes that are required in order for the manuscript to be considered scientific. The research design, hypotheses, and methodology should be clearly stated in the first section of the paper as well.
Response 1: Point well taken. The research design, hypothesis and methodology are now incorporated as separate sections are mentioned as well in the abstract.
The abstract could be restructured for a better understanding of the topic. Starting from the context and significance of the research, followed by the aims and methodology of the research, and concluding with the results and possible contribution of the research...Also, the scientific methodology used for the research should be presented in the abstract, as well as in the text. Response 2: The abstract now refers to the Please see the revised structure which addresses your concerns which include .
The connection between the text and the previous and present similar research should be elaborated more. The paper should consider a deeper background theoretical research in the domain of reconstruction or renewal and regeneration of architectural and cultural heritage, i.e. the projects that raise ecological awareness ...
Response 3: Indeed, the literature was limited, and we put great effort in reviewing existing literature in the areas of permaculture nature-based Solutions and participatory projects. Nevertheless, the reconstruction or renewal and regeneration of architectural and cultural heritage formally belongs in the upcoming mid-term stage. This type of inquiry did take place in the context of the class I teach where the DDRC was given as a case study, and a few very good student projects came out of it, looking into natural and artificial lighting proposals, incorporation of the site into a network of public spaces, etc. Also, the projects are written in Greek so it would be difficult to make use of them, but they will form part of the body of work that will be consulted for future steps.
Figure 3 is not readable, due to the small text size.
Response 4: Figure 3 has been replaced with a higher analysis one. I hope it works now.
There are several blank pages in the text (or part of the pages) pages 5 and 12.
Response 5: This problem has been fixed
The paper lacks a discussion of the results (related to the previous and present similar studies) as well as the overall conclusion of the research.
Response 6: The last part entitled results and further reflections has been extensively revised and we put the more effort in addressing the paper and especially this part is an assessment.
Round 2
Reviewer 2 Report
Thank you for submitting the revised version. The manuscript is somewhat improved than previous version.
What is difference between Background and Introduction here? I think it all should be under the headings of Background.
What does Figure 1 has importance in the manuscript?
Figure 1 is not required on Page 9.
Author Response
Thank you for submitting the revised version. The manuscript is somewhat improved than previous version.
What is difference between Background and Introduction here? I think it all should be under the headings of Background.
What does Figure 1 has importance in the manuscript?
Figure 1 is not required on Page 9.
Response to Reviewer 2 Comments
The two sections are now merged into one section which is entitled background. In the comment about Figure 1 it was somewhat unclear to us which was the image you referred to as redundant, as the Image number and the page number did not match, but we assumed that the comment was about the bird's eye view of the site. That image has been removed.
Finally, extensive further editing has taken place.
Reviewer 3 Report
Dear authors, thank you for the changes as well as answers to the comments.
Author Response
Response to Reviewer 3 Comments
Extensive editing has taken place which we believe further improves the aspects deemed nedded further improvement (context, theoretical background, research design, questions, hypotheses, methods, citations and references). In addition the closing section has been further developed with emphasis on the project's novel contributions, and on aspects that need attention for the project's viability.